# Less Binge Eating and Loss of Control over Eating Are Associated with Greater Levels of Mindfulness: Identifying Patterns in Postmenopausal Women with Obesity

**DOI:** 10.3390/bs9040036

**Published:** 2019-04-08

**Authors:** Veronica M. Smith, Radhika V. Seimon, Rebecca A. Harris, Amanda Sainsbury, Felipe Q. da Luz

**Affiliations:** The Boden Institute of Obesity, Nutrition, Exercise & Eating Disorders, Faculty of Medicine and Health, Charles Perkins Centre, The University of Sydney, Camperdown, NSW 2050, Australia; veronica.smith@sydney.edu.au (V.M.S.); radhika.seimon@sydney.edu.au (R.V.S.); rebecca.harris@sydney.edu.au (R.A.H.)

**Keywords:** mindfulness, obesity, eating behavior, eating disorders, binge eating, overeating, loss of control over eating, postmenopausal women

## Abstract

Obesity is a public health concern resulting in widespread personal, social, and economic burden. Many individuals with obesity report feeling unable to stop eating or to control their food intake (i.e., a *loss of control over eating*) despite their best efforts. Experiencing loss of control over eating predicts further eating pathology and is a key feature of binge eating. Mindfulness (i.e., awareness and acceptance of current thoughts, feelings, sensations, and surrounding events) has emerged as a potential strategy to treat such eating disorder behaviors, but it is not known whether there is merit in investigating this strategy to address binge eating in postmenopausal women with obesity. Thus, this study aimed to examine the relationships between binge eating and mindfulness in postmenopausal women with obesity seeking weight loss treatment. Participants (*n* = 101) were assessed with the Eating Disorder Examination Questionnaire, the Loss of Control over Eating Scale, the Five-Facet Mindfulness Questionnaire, and the Langer Mindfulness Scale. Participants´ overall scores on both mindfulness scales were significantly and negatively correlated with binge eating frequency or the severity of loss of control over eating. Moreover, participants who reported fewer binge eating episodes were significantly more mindful than those who reported greater frequencies of binge eating episodes within the past 28 days. These findings suggest a merit in investigating the use of mindfulness-based therapies to treat binge eating in postmenopausal women with obesity.

## 1. Introduction

Obesity is a major public health concern associated with a range of health complications resulting in high social and economic burden [1]. Individuals with overweight or obesity are more likely than people without overweight or obesity to recurrently engage in episodes of eating that are considered to be more than a normal amount of food (i.e., binge eating) [2], which can contribute to an individual’s inability to lose weight or to maintain a lower weight after weight loss [3]. Epidemiological data suggest a growing prevalence of individuals with obesity and comorbid eating disorder behaviors, such as binge eating [4]. Binge eating behaviors are associated with an impaired quality of life [4]; indeed, both eating-related and general psychopathologies are more pronounced in individuals with obesity who binge eat compared to those who do not binge eat [2]. In addition to binge eating, many individuals with obesity report feeling unable to stop eating or to control how much they are eating despite their best intentions [3]. This experience is clinically referred to as a *loss of control over eating* and is a key feature of binge eating in individuals with eating disorders [5]. Experiencing loss of control over eating predicts distress [6], impairment in psychosocial functioning [6], and global eating pathology [7] and is suggested to be a stronger predictor of psychological distress than the amount of food consumed [6,8]. Thus, the sense of loss of control over eating is included in the diagnostic criteria for binge eating disorder [5]. Binge eating is categorized into two subgroups to assist in capturing the full spectrum of this pathology: (1) objective binge eating, that is, when loss of control over eating is coupled with consuming what is considered to be a large amount of food; and (2) subjective binge eating, that is, when loss of control over eating is experienced but the amount consumed is not large [8].

Although both loss of control over eating and binge eating are psychological drivers of overeating that independently contribute to obesity, interventions for overweight and obesity rarely address such psychological factors; instead, they routinely focus on diet and physical activity [3]. Evidence suggests that individuals with eating disorders and comorbid overweight or obesity are at greater risk of health issues than individuals presenting with just one of these conditions [9]. Therefore, treatments for overweight and obesity that also address psychological factors contributing to loss of control over eating and binge eating are necessary to assist with weight loss treatment efficacy.

Potential strategies to reduce binge eating and associated behaviors are mindfulness-based interventions [10,11]. Mindfulness involves present awareness and acceptance (nonjudgement) of internal and external events (e.g., thoughts, feelings, sensations, environmental experiences), and has been associated with improved psychological well-being in clinical and nonclinical populations [10,11,12]. Hence, mindfulness interventions have also been suggested as potential treatment strategies for people with eating disorder behaviors and comorbid overweight or obesity [3,13,14,15]. The efficacy of mindfulness-based treatment was recently demonstrated in a clinical trial in which 194 men and women with obesity engaged in a diet and exercise program for five and a half months with or without the inclusion of mindfulness training [3]. Six months after completion of the intervention, individuals who had been randomized to the mindfulness treatment experienced less frequent loss of control over eating and had lost more weight compared with those who had been randomized to the control condition (i.e., no mindfulness training) [3]. Another clinical trial randomized 140 men and women with obesity who also met the diagnostic criteria for binge eating disorder to one of three conditions: a mindfulness-based treatment aiming to cultivate awareness of hunger and satisfaction, a cognitive-behavioral and educational-based treatment, and a wait-list control [13]. Out of the 92 participants who completed the 12-session treatments, those who received mindfulness-based treatment had significantly less binge eating episodes compared with those who received the cognitive-behavioral treatment and those on a wait-list control [13]. Furthermore, 68% of individuals in the mindfulness-based treatment no longer met the diagnostic criteria for binge eating disorder at four months post-treatment, compared with 46% of individuals receiving the alternative treatment and 36% of wait-list control participants. Despite differences in binge eating, participants in both treatment groups lost equivalent amounts of weight, suggesting that weight loss was unrelated to the observed reduction in binge eating in this study [13]. A recent meta-analysis examining the effects of mindfulness on health behaviors in individuals with overweight or obesity corroborated these findings [14]. Specifically, the aggregation of data across 12 randomized controlled trials revealed that mindfulness training significantly reduced binge eating but did not influence weight loss [14]. However, the authors of that meta-analysis noted important limitations within the reviewed body of literature, including small samples sizes and selection bias [14]. Furthermore, the above-reviewed studies included men and women aged 18 years and over [3,13,14], thus limiting the generalizability of these findings to specific populations, such as postmenopausal women, a population in which eating disorders appear common [16].

The abovementioned intervention studies are corroborated by studies suggesting an association between higher levels of mindfulness and lower eating pathology [17,18,19,20,21]. One of these publications [19] involved a longitudinal study with 300 female undergraduate students in which mindfulness was defined using a measure that examines an individual’s total mindfulness and five distinct facets of mindfulness (based on the constructs of the Five-Facet Mindfulness Questionnaire: *observe*, *describe*, *act with awareness*, *nonjudgement*, and *nonreact* [22]). Results revealed that greater levels of the facet named *nonreact* (i.e., higher nonreactivity to inner experience) predicted lower binge eating six months later [19]. This finding is consistent with previous research also examining female undergraduate students indicating that the facets of *act with awareness*, *nonjudgement*, and *nonreact* were negatively associated with eating pathologies [18]. Additionally, research with young women who were receiving formal treatment for eating disorders found that those with higher levels of certain aspects of mindfulness (e.g., awareness, acceptance) upon admission displayed lower eating disorder symptomology [21]. Furthermore, the women who reported the greatest increase in mindfulness throughout their treatment experienced the best treatment outcomes [21]. Together, these findings suggest that greater levels of mindfulness are associated with lower eating pathology among young women. However, the homogenous samples used within these studies limit the generalizability of these findings to other age groups (e.g., postmenopausal women).

While eating disorders are most prevalent in young women [23], up to 15.3% of postmenopausal women have reportedly met diagnostic criteria for an eating disorder [24]. Moreover, this figure may not accurately reflect the number of postmenopausal women who have eating disorders, as many women within this demographic remain undiagnosed. For instance, recent data [25] showed that middle-aged women with disordered eating behaviors predominantly exhibit binge eating behaviors that may or may not reach diagnostic levels, rather than “classical” eating disorder presentations (i.e., anorexia nervosa, bulimia nervosa, and binge eating disorder). Thus, their symptoms are more likely to remain undetected by clinicians who are less familiar with disordered eating behaviors that do not reach diagnostic criteria (i.e., other specified and unspecified feeding and eating disorders [5]). Despite the high prevalence of eating disorders in older women [24] and the understanding that eating disorder prevalence within this population has increased remarkably over the last decade [25], postmenopausal women remain largely understudied. Thus, the current study aims to explore the relationship between mindfulness and binge eating in postmenopausal women with obesity. In accordance with the literature suggesting that mindfulness-based techniques are an effective treatment strategy to reduce disordered eating behaviors and research demonstrating that higher levels of mindfulness are associated with lower eating pathology, it was predicted that higher levels of mindfulness would be associated with lower levels of binge eating and loss of control over eating.

## 2. Materials and Methods

### 2.1. Participants

Participants were 101 women aged 45–65 years (mean ± SEM = 58.0 ± 0.4 years) with obesity (mean ± SEM = 34.5 ± 0.2 kg/m^2^; 91.3 ± 0.9 kg) who were at least five years postmenopausal. Participants were enrolled in the TEMPO Diet Trial (**T**ype of **E**nergy **M**anipulation for **P**romoting optimum metabolic health and body composition in **O**besity, Australian New Zealand Clinical Trials Registry Number 12612000651886), a randomized controlled trial investigating the effects of weight loss via severe versus moderate dietary energy restriction on a range of physiological and psychological variables. Participant eligibility was described in full elsewhere [26]. In brief, eligible participants were ambulatory, sedentary (<60 min of physical activity per week), and had been weight stable for at least six months (±2 kg). Participants were ineligible if they had atypical thyroid function; loose metal in their body (which could affect magnetic resonance imaging, which was used for some outcome measures in the trial); diabetes; osteoporosis; alcohol or drug dependencies; currently smoked; or had taken medication within the last three years that affected bone mass, heart rate, body composition or appetite.

### 2.2. Measures

Participants completed a total of four questionnaires for this study: two questionnaires measuring eating disorder behaviors (the Eating Disorder Examination Questionnaire, EDE-Q, and the Loss of Control over Eating Scale, LOCES) and two questionnaires assessing mindfulness (the Five-Facet Mindfulness Questionnaire, FFMQ, and the Langer Mindfulness Scale, LMS). These questionnaires were administered two weeks prior to commencement of the intervention during a face-to-face appointment at our clinical research facility (for more details, refer to our previous publication [26]). Data from the LOCES were only available for 88 participants, as we started administering this scale after data collection had begun.

#### 2.2.1. Eating Disorder Examination Questionnaire (EDE-Q)

The EDE-Q, a modified 28-item version of the original Eating Disorder Examination, is a commonly used self-report questionnaire assessing eating disorder symptomology that displays high test–retest reliability and internal consistency [27]. Some minor modifications were made to the original EDE-Q to reflect the characteristics of the current sample, which are described in full elsewhere [26]. A noteworthy change was the addition of a question derived from a previous study [28] designed to capture subjective binge eating episodes: “Have you had other episodes of eating in which you have had a sense of having lost control and eaten more than you would like, but have not eaten a very large amount of food given the situation? Over the past 28 days how many days approximately would this have happened?” Participants also responded to standard questions such as: “Over the past 28 days, how many times have you eaten what other people would regard as an unusually large amount of food? … On how many of these times did you have a sense of having lost control over your eating?” Participants´ answers to these questions generated frequencies of subjective binge eating episodes and objective binge eating episodes, respectively. These frequencies were then summed together to produce a total score of binge eating episodes for each participant. The total score of binge eating episodes was used in all analyses, in light of the abovementioned research suggesting that the perception of loss of control over eating is a stronger predictor of psychological distress than the amount of food consumed [6].

#### 2.2.2. Loss of Control over Eating Scale (LOCES)

The LOCES is a 24-item self-report scale used in both clinical and nonclinical settings to assess how frequently and severely individuals feel they lose control over their ability to regulate their food intake [8]. While the EDE-Q we used asks a single question (outlined above) that distinguishes between people who *do* experience loss of control over eating versus those who *do not*, the LOCES captures different gradations of symptom severity. Thus, the LOCES provides additional information that can distinguish milder eating pathologies from more severe eating pathologies [8]. The total score for this measure ranges from a minimum of 24 (indicating that the participant does not experience loss of control over eating) to a maximum of 120 (indicating that the participant reports frequent and severe loss of control over eating).

#### 2.2.3. Five-Facet Mindfulness Questionnaire (FFMQ)

The FFMQ is a validated and commonly used psychological tool with 39 items that measures the construct of mindfulness [22,29]. The FFMQ measures five different components of mindfulness: *observe*, *describe*, *act with awareness*, *nonjudgement*, and *nonreact*. *Observe* is a type of awareness referring to an individual’s attendance to internal and external experiences (i.e., sensations, thoughts, emotions, and sounds) [22]. *Describe* refers to an individual’s inclination to describe these types of experiences in words; for example, an individual with a regular tendency to express their experiences in words would be rated as having higher levels of *describe* [30]. *Act with awareness* describes an individual’s ability to solely focus their attention on a current activity [30]. This is in contrast to acting without conscious awareness (i.e., being on “automatic pilot”). *Nonjudgement* refers to how negatively an individual evaluates inner experiences [22]. For example, individuals with low *nonjudgement* would rate themselves negatively, whereas individuals with high *nonjudgement* would rate themselves positively. Finally, the component *nonreact* describes an individual’s ability to experience thoughts and feelings without reacting towards them [22].

Scores from these components are summed to produce an overall score that infers an individual’s total mindfulness; the minimum possible score is 39, and the maximum is 195. In addition to the total FFMQ score, we also assessed a four-facet model of the FFMQ, with all facets except for *observe*, as this facet has been identified as a problematic measure [22,31]. Research has shown that this component specifically lacks validity when employed with people without meditation experience, suggesting that this measure may not produce uniform measurements when implemented with people who do or do not meditate [22,31]. The current study accounted for this discrepancy by running all statistical analyses considering both the FFMQ and the FFMQ (four-facet, excluding the *observe* component of the FFMQ). The minimum possible score for the FFMQ (four-facet) is 31, and the maximum is 155.

#### 2.2.4. Langer Mindfulness Scale (LMS)

The LMS was employed in addition to the FFMQ to assist with comprehensive measurement of mindfulness. This 14-item instrument was developed to capture Langer’s conceptualization of mindfulness: a state of mental functioning through which individuals are consciously aware of their surroundings and their engagement with the environment [32]. The LMS measures three components of mindfulness: *engagement* (i.e., an individual’s awareness of and interaction with his or her environment), *novelty seeking* (i.e., having an open and curious orientation to the environment), and *novelty producing* (i.e., an individual’s tendency to create new categories or information rather than rely on what he or she previously understands about a topic) [32]. For example, people with higher *novelty producing* scores are likely to create new mental associations (i.e., connections between one object or idea and another object or idea) when presented with situations in which they already possess some information. These three components of the LMS are summed together to produce a total score, whereby a higher score indicates higher levels of overall mindfulness. The minimum possible score on the LMS is 14, and the maximum is 98.

### 2.3. Statistical Analysis

Visual inspection of Normal Q–Q Plots and histograms revealed that all scores on the FFMQ and FFMQ (four-facet) were normally distributed. However, assumptions of normality were not met for scores on the LMS; the total score and all components of the LMS were negatively skewed, thus, participants’ scores on the LMS were generally high. Additionally, scores on the EDE-Q and LOCES were positively skewed; thus, participants’ scores on both of these measures were generally low. Therefore, nonparametric tests (Spearman’s rank-order correlations) were employed to investigate the relationships between mindfulness (as measured by the FFMQ, FFMQ (four-facet), and LMS) with the total frequency of binge eating episodes (from the EDE-Q) and severity of loss of control over eating (from the LOCES).

In addition to correlation analyses, participants were divided into two groups based on their total frequency of binge eating episodes, and the levels of mindfulness in the two groups were compared using independent *t*-tests. Participants were divided in two different ways. First, participants were divided into those who reported *no binge eating episodes* over the last 28 days, and those who reported *one or more* binge eating episodes over the last 28 days. This method of dividing participants was chosen so that differences between individuals who did or did not engage in binge eating could be investigated. Second, participants were divided into those who reported *three or fewer* binge eating episodes within the last 28 days, and those who reported *four or more* binge eating episodes. These thresholds were chosen based on the criterion for binge eating disorder from the Diagnostic and Statistical Manual of Mental Disorders (5th edition) [5], which states that binge eating episodes must occur at least once a week to reach the criterion for a clinical diagnosis. Hence, participants in the current study who reported three or fewer binge eating episodes within the last 28 days (i.e., an average of fewer than one per week) were classified as having *subclinical* levels of binge eating, and those who reported four or more binge eating episodes within the last 28 days (i.e., an average of one or more per week) were classified as having *clinical* levels of binge eating.

## 3. Results

### 3.1. Correlations between Mindfulness and Binge Eating or Loss of Control over Eating

As shown in Table 1, both binge eating and loss of control over eating were significantly negatively correlated with mindfulness. Indeed, both the total scores on the FFMQ and FFMQ (four-facet) were significantly negatively correlated with the total number of binge eating episodes experienced over the last 28-day period, as well as with perceived loss of control over eating. The total score on the LMS was significantly negatively correlated with the total number of binge eating episodes, albeit not with loss of control over eating. Regarding components of the FFMQ, significant negative correlations emerged between participants’ level of *act with awareness*, *nonjudgement*, and *nonreact* with the total number of binge eating episodes and loss of control over eating. Negative correlations also emerged between *describe* and binge eating episodes. In contrast, the *observe* component of the FFMQ was significantly and positively correlated with loss of control over eating. Regarding components of the LMS, *engagement* was significantly negatively correlated with both the total number of binge eating episodes, as well as with the perception of loss of control over eating. No other statistically significant relationships emerged (Table 1).

### 3.2. Comparison of Mindfulness in Individuals with Lower or Higher Levels of Binge Eating

As shown in Table 2 and Table 3, participants with lower levels of binge eating had significantly higher levels of mindfulness, as shown by the total scores for the FFMQ and FFMQ (four-facet), as well as their component of *nonjudgement* and the LMS component of *engagement*. The same pattern was observed within the FFMQ component of *describe* when comparing subclinical to clinical levels of binge eating (Table 3) but not when comparing zero versus one or more binge eating episodes (Table 2).

## 4. Discussion

This study comprehensively examined the relationships between mindfulness and binge eating in postmenopausal women with obesity. In line with our hypotheses, mindfulness was negatively associated with the total number of binge eating episodes within the last 28 days, as well as with participants’ perceived loss of control over eating. Furthermore, regardless of how data were divided (zero vs. one or more binge eating episodes, or three or fewer vs. four or more binge eating episodes in the past 28 days), participants who reported lower levels of binge eating had significantly higher mindfulness. Collectively, these findings show that higher levels of mindfulness are associated with less binge eating frequency and loss of control over eating in postmenopausal women with obesity. While total mindfulness scores allow for general interpretations to be made, exploring the relationships between binge eating behaviors and each component of mindfulness individually allowed for more detailed analyses and interpretations. These analyses revealed that participants with higher levels of the mindfulness components of *nonjudgement* (from the FFMQ) and *engagement* (from the LMS) were consistently less likely to report binge eating behaviors. Other mindfulness components from the FFMQ that were significantly associated with binge eating or loss of control over eating, albeit not consistently throughout all analyses, were *observe*, *describe*, *act with awareness*, and *nonreact*.

It is of interest that higher levels of *nonjudgement* were associated with lower levels of binge eating and loss of control over eating. Corroborating previous knowledge in undergraduate women [18], the current findings show that postmenopausal women with obesity who did not engage in binge eating behaviors, or who engaged in subclinical versus clinical levels of binge eating, displayed significantly higher levels of *nonjudgement* compared to those who did. Taken together, these findings suggest that postmenopausal women with obesity who do not evaluate inner experiences negatively are less likely to engage in binge eating behaviors.

*Engagement* is a component of the LMS that measures an individual’s capacity to actively attend to and be aware of the current environment and him- or herself. Consistent with previous research indicating that higher internal awareness is associated with less disordered eating [21], and that increasing internal attentiveness using mindfulness-based interventions can reduce habitual responses such as overeating [13,17], the current study found that participants with higher levels of mindfulness exhibited significantly less binge eating episodes compared with those with lower levels of mindfulness. These findings extend upon the current body of knowledge by indicating that cultivating engagement and awareness in the present moment may also be an important consideration for postmenopausal women with obesity.

Two other noteworthy components of the FFMQ in this study were *describe* and *nonreact*. Here, we found that participants with subclinical versus clinical levels of binge eating had higher levels of the *describe* component of the FFMQ. This suggests that participants who are more likely to express their experiences (e.g., sensations, thoughts, and emotions) in words are less likely to exhibit clinical levels of binge eating. However, while significant in some analyses, these associations did not occur consistently, and thus, the *describe* component of mindfulness may not be as critical as *nonjudgement* and *engagement* when considering binge eating behaviors specifically. Similar to *describe*, the observed relationships between eating pathology and levels of *nonreact*—referring to an individual’s ability to self-regulate and refrain from outwardly reacting to experiences—were inconsistent across analyses. While levels of *nonreact* were significantly negatively associated with the total number of binge eating episodes and perceived loss of control over eating, there was no difference in *nonreact* between participants with zero or subclinical levels of binge eating and those with some or clinical levels of binge eating, respectively. Taken together, these findings suggest that there may be merit in investigating the *describe* and *nonreact* aspects of mindfulness in future studies, but that these may not be fundamental characteristics associated with binge eating behaviors in postmenopausal women with obesity.

Finally, some contrasting findings emerged regarding the *observe* measure of the FFMQ; specifically, this component was significantly *positively* correlated with binge eating behaviors. However, substantial problems regarding the validity of the *observe* measure of the FFMQ have been reported elsewhere [22,31], as outlined in Section 2.2.3, and hence, the results from this measure that are in contrast to the rest of the findings from this study may not be relevant.

Limitations must be considered when interpreting these data. Due to the correlational nature of the present study design, it is plausible that other unmeasured third variable factors may have mediated the relationship between mindfulness and binge eating behaviors in our study. Also, the direction of causality cannot be elucidated from our study—that is, for instance, whether low mindfulness caused binge eating behaviors or vice versa, if at all. Finally, the differential correlations observed within the mindfulness subscales are of unclear significance since no factor analysis was employed to determine the extent to which intercorrelation existed between variables. However, the current study also has several strengths. Firstly, the construct of mindfulness was precisely and comprehensively defined; indeed, two interrelated and validated measures were used to assess this complex construct. Adopting these measures allowed for an overview of mindfulness and also provided specific information regarding individual components of mindfulness. Furthermore, the study controlled for concerns regarding the *observe* measure, as mentioned in Section 2.2.3 (i.e., that people who meditate or do not meditate may have different outcomes on this construct due to a training effect). The resultant four-facet FFMQ (the FFMQ without the *observe* component), as well as the FFMQ, were both assessed using correlations and *t*-tests, with results across both models being comparable, thereby enhancing confidence in the results. An additional strength of this study is that data were divided and analyzed in two distinct ways, allowing for thorough investigation. Indeed, regardless of how the data were divided or analyzed, results were largely cohesive, providing further validation of these findings and statistical methods. Another study strength was the medium to large sample size recruited. This is of particular interest since limited sample sizes are common within this research field and may affect statistical rigor [12,14]. Finally, the benefits of mindfulness-based treatments are generally assessed immediately or up to six months postintervention [11]. Thus, the current study contributes to the literature by assessing relationships between *stable* levels of mindfulness and how these general dispositions relate to binge eating behaviors.

## 5. Conclusions

This study showed that mindfulness was negatively associated with binge eating frequency and the severity of loss of control over eating in a sample of postmenopausal women with obesity. Additionally, we found that postmenopausal women with obesity without binge eating, or with subclinical levels of binge eating, reported higher characteristics of mindfulness—notably, the *nonjudgement* component of the FFMQ and the *engagement* component from the LMS—in comparison with those with higher or clinical levels of binge eating. These results suggest there is merit in investigating the use of mindfulness-based therapies to treat binge eating in postmenopausal women with obesity.

## Figures and Tables

**Table 1 behavsci-09-00036-t001:** Correlations between mindfulness and binge eating or loss of control over eating.

Mindfulness Scale/Questionnaire and Components	Binge Eating	Loss of Control over Eating
**Five-Facet Mindfulness Questionnaire (FFMQ)**		
Total FFMQ	**−0.270 ***	**−0.305 ****
Total FFMQ (four-facet)	**−0.349 ****	**−0.389 ****
Observe	0.130	**0.241 ***
Describe	**−0.205 ***	−0.131
Act with Awareness	**−0.215 ***	**−0.292 ****
Nonjudgement	**−0.434 ****	**−0.446 ****
Nonreact	**−0.207 ***	**−0.375 ****
**Langer Mindfulness Scale (LMS)**		
Total LMS	**−0.218 ***	−0.192
Engagement	**−0.342 ****	**−0.422 ****
Novelty Seeking	−0.089	−0.084
Novelty Producing	−0.062	0.060

FFMQ (four-facet) = the following four items from the FFMQ: describe, act with awareness, nonjudgement, and nonreact. Binge eating = total number of objective and subjective binge eating episodes in the past 28 days. * *p* < 0.05. ** *p* < 0.001.

**Table 2 behavsci-09-00036-t002:** Levels of mindfulness in participants who reported zero versus one or more binge eating episodes over the last 28 days.

Mindfulness Scale/Questionnaire and Components	Number of Binge Eating Episodes				Mean Difference [95% Confidence Interval]	
	Zero		One or More			
	Mean	SD	Mean	SD		*p*-Value
**Five-Facet Mindfulness Questionnaire (FFMQ)**	(*n* = 33)		(*n* = 64)			
Total FFMQ	145.45	18.53	137.14	16.63	**8.31 [0.96, 15.67]**	**0.027 ***
Total FFMQ (four-facet)	119.91	16.87	109.78	15.24	**10.13 [3.40, 16.85]**	**0.004 ***
Observe	25.55	5.53	27.36	4.57	−1.81 [−3.91, 0.28]	0.088
Describe	31.76	5.80	29.39	5.95	2.37 [−0.14, 4.88]	0.064
Act with Awareness	30.03	5.32	28.13	4.84	1.91 [−0.23, 4.04]	0.079
Nonjudgement	33.82	5.48	29.08	5.43	**4.74 [2.42, 7.06]**	**<0.001 ****
Nonreact	24.30	5.16	23.19	4.24	1.12 [−0.83, 3.06]	0.258
**Langer Mindfulness Scale (LMS)**	(*n* = 33)		(*n* = 64)			
Total LMS	78.03	11.79	74.88	7.17	3.15 [−0.67, 6.98]	0.105
Engagement	23.94	3.19	21.75	3.29	**2.19 [0.80, 3.58]**	**0.002 ***
Novelty Seeking	29.88	4.04	29.22	3.17	0.66 [−0.82, 2.14]	0.379
Novelty Producing	24.21	5.89	23.91	3.20	0.31 [−1.92, 2.53]	0.741

FFMQ (four-facet) = the following four items from the FFMQ: describe, act with awareness, nonjudgement, and nonreact. * *p* < 0.05. ** *p* < 0.001.

**Table 3 behavsci-09-00036-t003:** Levels of mindfulness in participants with subclinical versus clinical levels of binge eating (i.e., three or fewer vs. four or more episodes) over the last 28 days.

Mindfulness Scale/Questionnaire and Components	Number of Binge Eating Episodes				Mean Difference [95% Confidence Interval]	
	Three or Fewer (Subclinical)		Four or More (Clinical)			
	Mean	SD	Mean	SD		*p*-Value
**Five-Facet Mindfulness Questionnaire (FFMQ)**	(*n* = 54)		(*n* = 43)			
Total FFMQ	143.94	16.73	134.98	17.70	**8.96 [2.00, 15.93]**	**0.012 ***
Total FFMQ (four-facet)	117.52	15.52	107.84	16.15	**9.68 [3.27, 16.09]**	**0.003 ***
Observe	26.43	5.37	27.14	4.44	−0.71 [−2.73, 1.31]	0.485
Describe	31.37	5.60	28.72	6.17	**2.65 [0.27, 5.03]**	**0.029 ***
Act with Awareness	29.59	5.00	27.74	5.02	1.85 [−0.18, 3.88]	0.074
Nonjudgement	32.39	5.61	28.56	5.54	**3.83 [1.57, 6.09]**	**0.001 ****
Nonreact	24.17	4.87	24.17	4.87	1.36 [−0.49, 3.20]	0.149
**Langer Mindfulness Scale (LMS)**	(*n* = 54)		(*n* = 43)			
Total LMS	77.33	10.15	74.21	7.25	3.12 [−0.52, 6.77]	0.092
Engagement	23.33	3.17	21.44	3.43	**1.89 [0.56, 3.23]**	**0.006 ***
Novelty Seeking	29.72	3.57	29.09	3.37	0.63 [−0.78, 2.04]	0.379
Novelty Producing	24.28	5.00	23.67	3.18	0.60 [−1.14, 2.35]	0.493

FFMQ (four-facet) = the following four items from the FFMQ: describe, act with awareness, nonjudgement, and nonreact. * *p* < 0.05. ** *p* < 0.001.

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
