# Peer review of "Less Binge Eating and Loss of Control over Eating Are Associated with Greater Levels of Mindfulness: Identifying Patterns in Postmenopausal Women with Obesity"

_behavsci, 2019, doi:10.3390/bs9040036_

Round 1
Reviewer 1 Report
This is a very well written paper which shows an inverse correlation between binge eating/loss of control over eating and mindfulness in postmenopausal obese women. The correlation is well known; it is true that this study confirms this finding in the specific population of postmenopausal women. Some subscales of the mindfulness rating scales used in the study appear to be more involved in this correlation.
Overall the biggest problems with this study are the relative lack of originality and the fact that it does not report longitudinal data (acknowledged by the Authors).
I have some sympathy for creative titles and I am not suggesting the title should be changed in "Yet another study showing that loss of control over eating is negatively correlated to mindfulness", but "...identifying treatment ideas" in the title is with no doubts an overstatement. The idea of providing mindfulness based intervention for binge eating is hardly new as the introduction shows (and its results on weight loss very inconsistent).
In the methods section for some of the scales the number of items is not reported. The differential correlation of mindfulness subscales with binge eating is of unclear significance given that there is no factor analysis on this sample confirming the multifactorial nature of the instruments
Author Response
Please view the uploaded document for our responses to Reviewer 1.

Reviewer 2 Report
The article presents very important data supporting the role of mindfulness improvement in obesity prevention and treatment. The paper is also well written and give a clear message.
I have suggestions that might be (but not necessary) addressed in the paper.
1) Is it possible to tests the relation between all assessed psychological factors and BMI in the sample?
2) a valuable addition to the results presented would be a comparison of binge eating, loss of control over eating and mindfulness indicators between postmenopausal women with and without obesity. Do Authors have such data?
Author Response
Please view the uploaded document to view our responses to Reviewer 2.
